# Quantitative Sensory Testing in Late-Onset ATTRv Presymptomatic Subjects: A Single Center Experience

**DOI:** 10.3390/biomedicines10112877

**Published:** 2022-11-10

**Authors:** Stefano Tozza, Daniele Severi, Giovanni Palumbo, Vincenzo Provitera, Lucia Ruggiero, Raffaele Dubbioso, Rosa Iodice, Maria Nolano, Fiore Manganelli

**Affiliations:** 1Department of Neuroscience, Reproductive Science and Odonstomatology, University “Federico II”, 80131 Naples, Italy; 2Neurology Department, Skin Biopsy Laboratory, Istituti Clinici Scientifici Maugeri IRCCS, 82037 Telese Terme, Italy

**Keywords:** amyloidosis, TTR, ATTRv, carrier, quantitative sensory testing

## Abstract

**Backgrounds** Hereditary transthyretin amyloidosis (ATTRv) presymptomatic subjects undergo multidisciplinary evaluation to detect, as early as possible, a subclinical involvement of multisystem disease. Quantitative sensory testing (QST) that investigates and discriminates the function of C, Aδ and Aβ fibers is included as an instrumental test to monitor nerve fiber function. The purpose of this study was to evaluate the role of QST in the context of the multidisciplinary evaluation in late onset carriers. **Methods** Four-teen presymptomatic (namely carriers) were enrolled. Subjects underwent thermal [cold and warm detection threshold (CDT, WDT), cold and heat pain (CP and HP)] and tactile QST in four body sites: foot dorsum, distal lateral leg, distal thigh, hand dorsum. **Results** Overall, presymptomatic subject showed a significant difference in all thermal QST findings compared to the control group. All subjects had at least one altered thermal QST finding; the sites more frequently altered were foot and leg, whilst the thermal modalities which were more frequently abnormal were CDT, WDT and CP. **Conclusions** Our study highlights the importance of performing thermal QST in subjects carrying TTR mutation, given the high frequency of abnormal findings. Notably, performing both innocuous and painful stimulation in foot and/or leg increases the chance of detecting nerve fiber dysfunction. Moreover, the investigation of the hand may provide useful information in monitoring disease progression before the Predicted Age of Disease Onset (PADO).

## 1. Introduction

Hereditary transthyretin amyloidosis (ATTRv) is a rare disease caused by mutation of the *transthyretin* (*TTR*) gene, characterized by extracellular deposition of amyloid fibrils in the peripheral nervous system (PNS) and the heart [1]. PNS involvement leads to a rapidly progressive and disabling sensory-motor axonal neuropathy [2]. The different involvement of sensory nerve fibers was demonstrated depending on mutation type [3]. The early onset phenotype (<50 years), typical of the endemic regions, is characterized by the predominant small fiber involvement, while the late onset phenotype (≥50 years), typical of non-endemic areas such as Italy [4], displays a progressive and prevalent large fiber damage with partial sparing of the smallest fibers [5,6].

In the last decade, different therapies (TTR tetramer stabilizer and gene silencers) were approved for treating ATTRv disease [7,8,9] and they have drastically changed disease natural history. Nevertheless, the available anti-amyloid therapies can be prescribed only in patients with demonstrated polyneuropathy. Therefore, asymptomatic relatives (e.g., siblings, offspring), that result in carrying a pathogenic variant in TTR after the genetic counselling [10], cannot access available therapy.

In the era of effective therapy, the scientific community is paying great attention to presymptomatic subjects (carriers), especially regarding the issue of when ATTRv carriers become “symptomatic” and need a specific treatment [11]. Early diagnosis of disease in carriers is crucial to initiate therapy as soon as possible and remains an important challenge for clinicians that follow ATTRv carriers. Recently, an expert consensus proposed minimum criteria to define a carrier as a “symptomatic” patient; in the absence of symptoms, a subject with two abnormal instrumental findings can be considered as symptomatic and needs to be treated [12].

A multidisciplinary evaluation (PNS, heart, kidney) is required to detect the involvement of active multisystemic disease in presymptomatic subjects. Quantitative sensory testing (QST) is proposed as one of the neurophysiological tests to study especially the smallest fiber’s function. Indeed, depending on the sensory stimulus (e.g., warm, cold, tactile), QST allows to study the function of the different fibers (e.g., fibers C, Aδ and Aβ, respectively). Previous studies already demonstrated that thermal QST is a useful method for documenting the small fiber dysfunction before any objective electrophysiological signs of PNS involvement in presymptomatic subjects, especially in those with early onset Val30Met variant [13,14].

The aim of our study is to evaluate the role of QST in the context of the multidisciplinary evaluation in presymptomatic subjects carrying a late onset phenotype variant in *TTR* gene. Moreover, the present work aims to establish which sensory stimuli and which site are more useful in demonstrating nerve fibers dysfunction.

## 2. Materials and Methods

We retrospectively evaluated the QST findings in the context of the multidisciplinary (neurological, cardiological and nephrological) evaluation that we used to perform in presymptomatic subjects in our third level neuromuscular center (University of Naples “Federico II”).

We included people carrying a pathogenic variant in *TTR* gene without any symptom and sign of ATTRv and with normal nerve conduction study (NCS). QST was performed according to standardized protocol [15] to study the function of C (warm stimuli), Aδ (cold stimuli) and Aβ (tactile stimuli) fibers. QST was assessed in four body sites: foot dorsum, distal lateral leg, distal thigh, hand dorsum on the non-dominant side.

In details, thermal QST was conducted through a thermal stimulator Thermal Sensory Analyzer II (TSA-2001, Medoc Ltd., Ramat Yishai, Israel) with the method of limits and using a thermal probe of 3 × 3 cm. The baseline temperature was 32 °C, with a lower cut-off temperature at 0 °C and upper cut-off temperature at 50 °C, and a ramp rate for all thermal stimuli was 1 °C/s. Cold detection threshold (CDT), warm detection threshold (WDT), cold pain (CP) and heat pain (HP) were assessed. Each threshold was expressed as the absolute difference in temperature from the baseline temperature (32 °C).

Tactile QST was performed by using a standardized set of calibrated monofilaments (Aesthesio Precision Tactile Sensory Evaluator, DanMic Global LLC, San Jose, CA 95124, USA) which exert forces between 0.008 g and 300 g. Moving stepwise from the thicker towards the thinner filament, tactile threshold (TT) was defined as the thinnest filament perceivable at least 5 times out of 10 [15]. Null stimuli were randomly applied during the test to evaluate subject reliability.

Reference QST thresholds were recorded in 55 healthy volunteers, showing no signs or symptoms of generalized or focal neuropathy, without any disease that can influence QST findings (e.g., diabetes, alcohol abuse). Age-matched Z-score [=(single patient X-healthy control mean)/healthy control SD] was calculated for each QST result and a Z-score greater/lower than ±2 was considered as abnormal value.

For the analysis, we collected the degree of CDT, WDT, CP, HP (°C), the weight of filament for TT (gr), and the frequency of abnormal results for each QST (≥+2 z-score). Moreover, we collected demographic data (age, gender, type of mutation), estimated the Predicted Age of Disease Onset (PADO) [12] and calculated the time to PADO (Time-to-PADO = PADO-age at evaluation).

### Statistical Analysis

Descriptive statistics were based on mean ± standard deviation in the case of numerical variables and on percentage in the case of categorical data. Since the variables had Gaussian distribution (verified through the Kolmogorov-Smirnov test), a Student’s T test was used to compare QST variables between carrier and control groups, between gender and between type of mutation, while a Chi-squared test was used to compare categorial variable (gender) between the two groups. To test if subjects closer to PADO had higher QST thresholds, ANOVA one-way analysis was performed between three groups: controls, ≤10 years to PADO group (*n* = 6; −3.5 ± 6.3; −15–4) and >10 years to PADO group (*n* = 8; 21.6 ± 8.1; 12–32). *p* values lower than 0.05 were deemed as statistically significant. Analyses and graphics were performed with statistical software IBM SPSS Statistic version 25 (SPSS Inc., Chicago, IL, USA).

## 3. Results

We recruited 14 presymptomatic subjects (belonging to 11 families) with no ATTRv symptoms or signs and with normal nerve conduction study. Demographic data and QST findings of ATTRv carrier and healthy controls were summarized in Table 1. Briefly, presymptomatic subjects showed a significant difference in all thermal QST findings compared to the control group. Conversely, no significant difference was found between the two groups in tactile QST results.

Analyzing the frequency of abnormal QST results (≥+2 z-score) in carrier cohort (Table 2), all subjects had at least one altered thermal QST finding (100% of abnormal results). In Figure 1, the z-score of thermal QST at leg site of all 14 patients is reported as an example.

The sites more frequently altered were foot (62.3%) and leg (62.3%), while less frequently involved were thigh (47.8%) and hand (44.9%). The sensory modality more frequently altered were CDT (74.1%), WDT (70.4%) and CP (68.6%), whilst less frequently involved was HP (48.1%) (Table 2).

ANOVA one-way analysis, to explore if QST results worsened with the PADO approaching, revealed that there was a statistically significant difference in all thermal QST findings between at least two groups (*p* < 0.05). On the other hand, no statistical differences were found in tactile QST data between groups (*p* > 0.05). The results of post-hoc Tukey’s HSD Test for multiple comparisons are graphically summarized in Figure 2.

Briefly, the post-hoc comparisons found that the mean value of all thermal QST findings were significantly different between the control group and the two carrier groups (> and ≤10 years to PADO). Conversely, the post-hoc analysis between the two carrier groups (> and ≤10 years to PADO) showed a significant mean difference (*p* < 0.05) of CDT and CP at foot, CP at leg and thigh, and WDT, CP and HP at hand site.

## 4. Discussion

ATTRv is a systemic disease involving especially PNS. The late-onset phenotype is characterized by signs and symptoms due to the predominant large fiber damage, while small nerve fiber involvement may remain for a long time at subclinical level. However, although NCS represents the gold standard for diagnosis of large fiber involvement, it can be completely unremarkable in an asymptomatic subject. Therefore, in the last years, to improve the capability in demonstrating PNS involvement in late onset phenotype, several tests exploring the smallest nerve fiber function were proposed to monitor carriers (e.g., skin biopsy, cutaneous silent period, QST) [13,14,16,17]. Among these tests, thermal QST allows for contemporary study and to discriminate the function of C and Aδ nerve fibers through the warm and cold stimulation, respectively.

In this study, we used the QST to further describe the involvement of nerve fibers in a late-onset ATTRv carrier cohort. Our results support the notion that the smallest fiber dysfunction was evident, since all thermal thresholds were higher compared to the control group. These findings support that C and Aδ fibers are precociously involved compared to the larger Aβ fibers (explored through tactile stimuli). Our results corroborate the morphological findings of sural nerve [18] and skin biopsy [16] in ATTRv carriers in which degeneration of unmyelinated (C fibers) and small myelinated (Aδ) fibers starts several years before the onset of symptoms, while Aβ fiber loss appears later in the natural history of disease.

In our cohort, all patients showed at least one abnormal threshold in one site. In the context of carrier’s multidisciplinary evaluation, QST allows us to always detect at least one abnormal test among those suggested by European consensus [12] and to consider carriers as “symptomatic” if another instrumental test results was altered.

The high frequency of abnormal QST findings, especially in the distal sites (foot and leg), appears to be in contrast with previous reports [13,14] that have applied this technique in ATTRv carriers and found a lower frequency of abnormal results. However, this work is the first study that applies both innocuous (CDT, WDT) and painful (CP, HP) thermal stimuli in four body sites in a cohort of carrier patients with late-onset phenotype. The high frequency might be related to the increased chance of detecting any type of dysfunction on four different body sites. Moreover, our results also seem different with respect to the frequency of abnormal epidermal innervation (64%) in presymptomatic through skin biopsy analysis [16]. However, the two techniques focused on different aspect of nerve fibers. Skin biopsy is a morphological evaluation of epidermal nerve fibers, whilst QST involves functional assessment of the smallest sensory fibers with the ability of discriminating C fibers from Aδ [19]. According to pathophysiological hypothesis, the deposition of TTR amyloid fibrils in Dorsal Root Ganglia in the presymptomatic stage might cause early neuronal dysfunction through several mechanisms as altered axon-glial cells crosstalk [20], membrane fluidity [21] and permeability [22]. The chronic dysfunction might ultimately determine cell death and peripheral axonal loss. Therefore, this supports the notion that nerve fibers in presymptomatic stage can appear still morphologically preserved but functionally impaired [16].

Moreover, this study surprisingly demonstrated that Aδ fibers were more frequently abnormal for both innocuous (74.1%) and painful (68.6%) cold stimulations compared to the C fibers that were altered more for innocuous (70.3%) respect to the painful (48.1%) warm stimulations. These results might support that in late-onset carriers the smallest myelinated fibers (Aδ) were more frequently involved compared to the unmyelinated C fibers. To support this, CP was the threshold more sensitive to detecting progression when subjects approached PADO, since in all explored sites CP were higher in < 10 years to PADO group (Figure 2). Interestingly, an in vitro study demonstrated that TTR aggregates influences the function of TRPM8 and Nav1.8 [23], detrimental channels for cold sensation and cold pain [24]. All together, these findings may support the importance of testing particularly the cold sensation through both innocuous and painful stimuli.

Lastly, the site more sensitive to detecting the progression due to approaching the PADO was the hand since three different sensory modalities (CDT, WDT, CP) out of four were different between the two groups (Figure 2).

## 5. Conclusions

In conclusion, this research, although presenting several limitations as a small cohort and the retrospective study design, is the first study focused on the role of QST in a group of carriers of late-onset ATTRv phenotype, including four sites and both innocuous and painful thermal stimuli. Our work suggests that thermal QST, despite the limits of the methodology due to its subjective nature, can be useful in multidisciplinary evaluation of carriers given the high frequency of abnormal findings. Notably, performing both innocuous and painful stimulation in foot and/or leg increases the chance of detecting nerve fiber dysfunction. Moreover, the investigation of the hand may provide useful information in monitoring disease progression before the PADO.

## Figures and Tables

**Figure 1 biomedicines-10-02877-f001:**
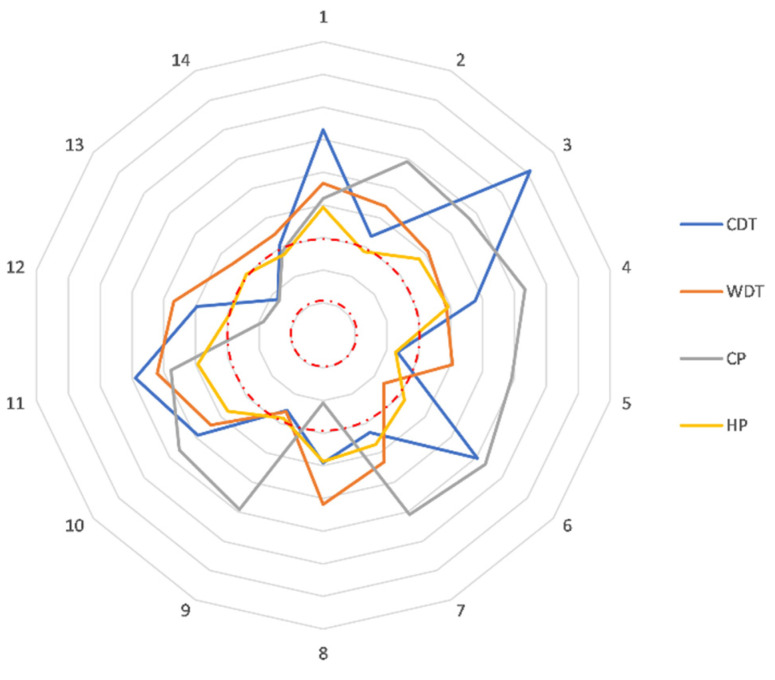
Radar representation of thermal QST at leg site. In the figure was represented the z-score value of all thermal modalities for each patient (each radius of the wheel). The red dotted circle lines represent the abnormal z-score value cut-off (the outer line is the +2 z-score, and the inner one is the −2 z-score).

**Figure 2 biomedicines-10-02877-f002:**
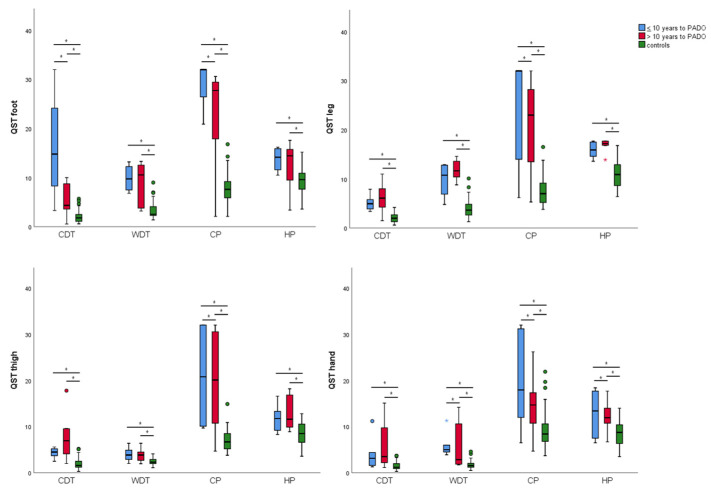
Comparison of QST finding between groups. Each panel represents a single stimulation site (foot, leg, thigh, hand). On the *y*-axis is reported the absolute temperature (°C) and on the *x*-axis is reported the different type of sensory stimulation (CDT = cold detection threshold; WDT = warm detection threshold; CP = cold pain; HP = heat pain). Different color denotes different group (in blue the group of patients with a time-to-PADO ≤10 years; in red the group of patients with a time-to-PADO > 10 years; in green the group of healthy controls). Asterisks (*) indicates significant *p*-value between groups (*p* < 0.05).

**Table 1 biomedicines-10-02877-t001:** Demographic and QST findings of ATTRv carriers and healthy controls.

	ATTRv Carriers (*n* = 14)	Healthy Controls (*n* = 55)	*p*-Value
**Age**	51 ± 11.2 (34–70)	50.8 ± 6.6 (37–61)	0.97
**Gender (M/F)**	11/3	23/22	0.70
**Mutation** Val30Met Phe64Leu	9/14 (64.3%) 5/14 (35.7%)	-	
**PADO**	61.2 ± 7.6 (40–72)	-	
**Time-to-PADO**	10.8 ± 14.7 (−15–32)	-	
**Thermal QST (°C)**			
Foot			
CDT	9.1 ± 8.5 (0.6–32)	2 ± 1.2 (0.6–5.7)	**0.016**
WDT	9.1 ± 3.8 (3.2–13.3)	3.3 ± 1.6 (1.4–9)	**<0.001**
CP	24.9 ± 8.9 (2.1–32)	7.8 ± 2.9 (2.1–16.8)	**<0.001**
HP	12.9 ± 4.1 (3.4–17.6)	9.2 ± 2.7 (3.6–15.2)	**<0.001**
Leg			
CDT	5.7 ± 2.4 (1.5–11)	2.1 ± 0.9 (0.6–4.2)	**<0.001**
WDT	10.9 ± 2.7 (4.8–14.6)	4 ± 1.8 (1.3–10.1)	**<0.001**
CP	22.4 ± 10.7 (5.3–32)	7.6 ± 3 (3.8–16.5)	**<0.001**
HP	16.5 ± 1.5 (13.6–17.7)	10.9 ± 2.8 (6.4–16.8)	**<0.001**
Thigh			
CDT	5.7 ± 2.4 (1.5–11)	1.9 ± 1.1 (0.3–5.2)	**0.002**
WDT	3.8 ± 1.4 (1.9–6.4)	2.4 ± 0.6 (1.1–4.1)	**0.002**
CP	20.3 ± 10.8 (4.7–32)	7 ± 2.2 (3.8–14.9)	**0.001**
HP	12.4 ± 3.4 (8.3–18.2)	8.4 ± 2.3 (3.6–12.8)	**0.001**
Hand			
CDT	5.1 ± 4.4 (1.2–15.1)	1.5 ± 0.8 (0.3–12.8)	**0.009**
WDT	5.8 ± 4.1 (1.7–14.2)	1.7 ± 0.7 (0.5–4.6)	**0.003**
CP	16.8 ± 8.6 (4.7–32)	9.2 ± 3.9 (3.7–21.9)	**0.008**
HP	12.4 ± 3.9 (6.5–18.4)	8.4 ± 2.3 (3.5–14)	**0.002**
**Tactile QST (gr)**			
Foot	1 ± 0.9 (0.07–4)	0.4 ± 0.4 (0.1–1.9)	0.147
Leg	1.6 ± 2.1 (0.16–8)	0.7 ± 0.5 (0.25–1.9)	0.153
Thigh	0.3 ± 0.2 (0.04–1)	0.4 ± 0.2 (0.1–1.1)	0.259
Hand	0.1 ± 0.1 (0.04–0.4)	0.2 ± 0.2 (0.1–1.1)	0.062

PADO = predicted of age disease onset; QST = quantitative sensory testing; CDT = cold detection threshold; WDT = warm detection threshold; CP = cold pain; HP = heat pain. Data of thermal QST are expressed in °C as the difference between the baseline probe temperature (32 °C) and the patient threshold. Data of tactile QST are expressed in gr. In bold are reported the significant *p*-value (<0.05).

**Table 2 biomedicines-10-02877-t002:** Abnormal frequency for each site and thermal stimulation.

	CDT	WDT	CP	HP	Tactile	Total
foot	83.3%	66.7%	91.7%	50.0%	23.1%	62.3%
leg	78.5%	85.7%	69.2%	64.3%	14.3%	62.3%
thigh	71.4%	50%	69.2%	42.9%	7.1%	47.8%
hand	64.2%	78.5%	46.1%	35.7%	0%	44.9%
total	74.1%	70.4%	68.6%	48.1%	9.1%	

CDT = cold detection threshold; WDT = warm detection threshold; CP = cold pain; HP = heat pain.

## Data Availability

Not applicable.

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
