# Peer review of "Quantitative Sensory Testing in Late-Onset ATTRv Presymptomatic Subjects: A Single Center Experience"

_biomedicines, 2022, doi:10.3390/biomedicines10112877_

Round 1

Reviewer 1 Report

This article entilted Quantitative sensory testing in late-onset ATTRv presymptomatic subjects: a single center experience. submited to Biomedicines journal is a retrospective orginal case-control study. Despite the clear limitations of this methodology it presents high scientific soundes. It is well witten and comprehensove. There are no ethical concerns. This study is very important to the field of amyloidosis research.   The presented results are precursor and may translate into a better assessment of the early stage of the disease.

Author Response

we thank the reviewer for the revision. We added on line 228 the limit of QST method since it was not reported in the previous manuscript.  

Reviewer 2 Report

The authors retrospectively evaluated the findings of quantitative sensory testing (QST) in 14 asymptomatic subjects with the mutation causing hereditary transthyretin (ATTRv) amyloidosis and compared to those in 55 healthy volunteers. Abnormal thermal QST findings were observed more or less in all asymptomatic subjects, indicating the utility of thermal QST for monitoring asymptomatic ATTRv carriers. 

This is an important study providing important insights into current knowledge on the management of ATTRv amyloidosis, particularly asymptomatic ATTRv carriers. Taking up the topics of ATTRv amyloidosis is timely because novel disease-modifying therapies, such as transthyretin stabilizers, RNA interfering agents, antisense oligonucleotides, and gene editing agents, now appear one after another. Therefore, this manuscript will attract broad range of readers. It is well written, and I do not have any critical comments.

Minor issues to strengthen this manuscript are raised as follows: 

1. Please reconfirm the use of abbreviations. For example, “PADO” in the abstract should be spelled out. 

2. Findings obtained from Figures 1 and 2 should be briefly explained in their legends although they are mentioned in the main text. 

3. The authors described pathological differences between early- and late-onset forms of ATTRv amyloidosis. As this issue has been demonstrated in an earlier study (Neurology 2004; 63: 129-38), it should be cited here. 

Author Response

The authors retrospectively evaluated the findings of quantitative sensory testing (QST) in 14 asymptomatic subjects with the mutation causing hereditary transthyretin (ATTRv) amyloidosis and compared to those in 55 healthy volunteers. Abnormal thermal QST findings were observed more or less in all asymptomatic subjects, indicating the utility of thermal QST for monitoring asymptomatic ATTRv carriers. 

This is an important study providing important insights into current knowledge on the management of ATTRv amyloidosis, particularly asymptomatic ATTRv carriers. Taking up the topics of ATTRv amyloidosis is timely because novel disease-modifying therapies, such as transthyretin stabilizers, RNA interfering agents, antisense oligonucleotides, and gene editing agents, now appear one after another. Therefore, this manuscript will attract broad range of readers. It is well written, and I do not have any critical comments.

Minor issues to strengthen this manuscript are raised as follows: 

Please reconfirm the use of abbreviations. For example, “PADO” in the abstract should be spelled out. 

Response: we spelled out the abbreviation PADO in the abstract (line 26).

Findings obtained from Figures 1 and 2 should be briefly explained in their legends although they are mentioned in the main text. 

Response: we added the caption of Figure 1 and 2 as requested.

The authors described pathological differences between early- and late-onset forms of ATTRv amyloidosis. As this issue has been demonstrated in an earlier study (Neurology 2004; 63: 129-38), it should be cited here. 

Response: we added the reference 5 at line 37.
